# A Simple and Efficient CRISPR Technique for Protein Tagging

**DOI:** 10.3390/cells9122618

**Published:** 2020-12-05

**Authors:** Fanning Zeng, Valerie Beck, Sven Schuierer, Isabelle Garnier, Carole Manneville, Claudia Agarinis, Lapo Morelli, Lisa Quinn, Judith Knehr, Guglielmo Roma, Frederic Bassilana, Mark Nash

**Affiliations:** Novartis Institutes for Biomedical Research, 4002 Basel, Switzerland; valerie.beck@novartis.com (V.B.); sven.schuierer@novartis.com (S.S.); isabelle.garnier@novartis.com (I.G.); carole.manneville@novartis.com (C.M.); claudia.agarinis@novartis.com (C.A.); lapo.morelli@novartis.com (L.M.); lisa.quinn@novartis.com (L.Q.); judith.knehr@novartis.com (J.K.); guglielmo.roma@novartis.com (G.R.); frederic.bassilana@novartis.com (F.B.); mark.nash@novartis.com (M.N.)

**Keywords:** CRISPR knock in, non-homologous end-joining (NHEJ), protein tagging

## Abstract

Genetic knock-in using homology-directed repair is an inefficient process, requiring the selection of few modified cells and hindering its application to primary cells. Here, we describe Homology independent gene Tagging (HiTag), a method to tag a protein of interest by CRISPR in up to 66% of transfected cells with one single electroporation. The technique has proven effective in various cell types and can be used to knock in a fluorescent protein for live cell imaging, to modify the cellular location of a target protein and to monitor the levels of a protein of interest by a luciferase assay in primary cells.

## 1. Introduction

Genome editing has been revolutionized by the discovery of the CRISPR-Cas9 system. Cas9 nuclease creates a double-strand DNA break (DSB) at a target site specified by the guide RNA (gRNA) sequence [1,2]. In the presence of a donor DNA template, gene tagging can be achieved through homology-directed repair (HDR), the efficiency of which is low and not applicable for all cell types. Compared with HDR, non-homologous end-joining (NHEJ) is more efficient and is active in both proliferating and post-mitotic cells. Indeed, DSBs are preferentially repaired via the NHEJ pathway [3]. Although commonly used for gene knock-out by causing insertion or deletion (indel) mutations, NHEJ repair has been demonstrated to be intrinsically accurate and can also be used to create gene knock-ins [4,5,6,7].

The application of CRISPR knock-in in cell biology is hampered by its low efficiency and laborious procedure. The aim of this study was to develop a CRISPR knock-in protocol that is: (1) highly efficient; (2) easy to use; and (3) applicable to cell lines and primary cells. We chose electroporation as the delivery method for the gRNA/Cas9 nuclease/donor DNA complex, as it offers better genome editing efficiency [8,9,10]. We opted to use dsDNA without any homology arms as the donor and hypothesized that if dsDNA donors are available in abundance when and where DSB happens, they could be incorporated into the genome via NHEJ with high degrees of accuracy. With this strategy, we can efficiently tag and test the target proteins within a short timeframe. We name it Homology independent gene Tagging (HiTag).

## 2. Materials and Methods

gRNAs. We used the Alt-R CRISPR-Cas9 System from Integrated DNA Technologies (IDT) (Leuven, Belgium). The gRNAs were designed using the CRISPOR program [11] (http://crispor.tefor.net/) and are listed in Appendix A. crRNA and tracrRNA were dissolved with Duplex buffer (IDT) to 169 µM each. They were mixed at the equal volume, and the final RNA concentration was adjusted to 2 µg/µL by adding Duplex buffer. The gRNA complex was formed by incubation at 95 °C for 5 min, followed by slowly cooling down to room temperature (RT), and then stored at −80 °C.

Cas9 nuclease. Cas9 proteins were mostly produced in-house. Briefly, NLS-SpyCAS9-NLS-His6 was expressed in E. coli strain Rosetta 2 (DE3) in pET28a+, using 2xYT media plus 50 µg/mL kanamycin overnight at 18 °C. Cas9 was then purified using Ni-NTA agarose (Qiagen, Germantown, MD, USA) capture/elution, followed by size exclusion chromatography on an S200 26/600 Superdex column (GE Lifesciences, Marlborough, MA, USA) using modified buffers [12] from Jinek et al. Then, Cas9 stocks (5.9 mg/mL) were aliquoted and stored at −80 °C. Commercially available Cas9 nuclease (IDT) can also be used, with no difference observed in our hands (data not shown). When the HiTag protocol was tested in HCT116 cells stably expressing Cas9, the knock-in efficiency was equivalent to that by Cas9 proteins (data not shown).

dsDNA donors. Oligonucleotides were synthesized by Microsynth with PAGE purification. The forward and reverse strands were dissolved in Duplex buffer to 20 µM each. To anneal the oligos, they were mixed at equal volume and incubated at 95 °C for 5 min and then allowed to cool down slowly to RT. The final concentration of the dsDNA stock is 10 µM. Unless stated otherwise, final concentrations of dsDNA in the transfection mixtures are 750 nM.

Transfection mixture. For each transfection, we prepared 20 µL of a cell/ribonucleoprotein/dsDNA mixture. Cells were trypsinized and washed once with phosphate-buffered saline (PBS) and then resuspended at the concentration of 5.4 × 106 cells per 100 µL buffer R (IDT). Ribonucleoprotein (RNP) was formed by incubating 4 µg Cas9 (18.4 pmol or 0.68 µL) and 2 µg gRNA (50 pmol or 1 µL) at RT for 10 min in a final volume of 5 µL Duplex buffer. Then, 400,000 cells (in 7.5 µL buffer R), 15 pmol dsDNA (1.5 µL) and 6 µL buffer R were added to make up the transfection mixtures (Appendix A).

For the knock-in of mScarlet, dsDNA donors were ordered as gBLOCK gene fragments (IDT). An amount of 500 ng DNA was dissolved in 5 µL buffer R (IDT). The constitutions of the transfection mixtures are listed in Appendix A.

Electroporation. The RNP and dsDNA templates were delivered into cells by electroporation with the Neon Transfection System (MPK5000) (ThermoFisher, Waltham, MA, USA). We used the 10 µL kit (MPK1096) (ThermoFisher, Waltham, MA, USA), and transfection protocols for individual cell types are listed in Appendix A. After transfection, cells were cultured in antibiotic-free medium for at least 24 h before replacing with normal growth medium.

Cell culture. All cells were maintained under a humidified atmosphere of 5% CO_2_ at 37 °C. A549, HCT116, HEK293, HeLa, SH-SY5Y and U-2 OS cell lines were purchased from ATCC and cultured according to the supplier’s recommendation. The primary human skeletal muscle cells (CC-2561, Lonza, Basel, Switzerland) were maintained in Skeletal Muscle Growth Medium (Lonza, CC-3246), supplemented with 20% fetal calf serum (FCS).

Schwann cells culture. Animal experiments were performed in accordance with the Swiss ordinance on animal experimentation, after approval by Kantonales Veterinäramt Basel-Stadt. Primary Schwann cells were isolated as described previously by Kaewkhaw et al. [13] Briefly, sciatic nerves were isolated from adult rats and the epineurium was stripped. The tissues were teased with sharp tweezers and digested with 0.05% collagenase digestion at 37 °C for 2 h. The homogenous cell suspension was obtained by dissociating the tissue with a glass Pasteur pipette and filtered through a 40 µm cell strainer. The cells were pelleted down and plated in poly-d-lysine-coated flasks. The cells were maintained in a custom-made D-valine DMEM medium (based on 10313039, l-Valine free, ThermoFisher, Waltham, MA, USA), supplemented with 10% FCS, B27, 10 mM Hepes, 2 mM glutamate and 5µM Forskolin. Once the cultures reach confluency, Schwann cells were dispatched by Dispase (Corning) digestion, and they can be sub-cultured for up to 10 passages. The purities of Schwann cells were at least 95% at the time of transfection, judged by immunostaining with Schwann cell markers.

Generation and maintenance of iPS cells. Neonatal human dermal fibroblasts (ThermoFisher) were used for reprogramming using Sendai virus with the help of the CytoTune-iPS reprogramming kit according to the standard protocol. Colonies with the hallmark of a pluripotent morphology were readily visible between days 17 and 20 after transduction. These were picked and sub-cloned multiple times on plates coated with Matrigel (BD Biosciences, San Jose, CA, USA) in mTeSR1 medium (STEMCELL Technologies) until Sendai virus RNA could no longer be detected and the morphology looked stable. Pluripotency was controlled by FACS analysis. Potential for differentiation into the three germ layers was approved by using the TaqMan hPSC Scorecard Panel (ThermoFisher) according to the supplier’s guideline. Karyotype analysis was performed by full-genome SNP analyses conducted with Life & Brain and showed no larger chromosomal aberrations. iPSCs were maintained on Matrigel-coated plates and grown in mTesR1.

Generation and differentiation of iNgn2 iPSCs. Human Ngn2 cDNA was synthesized using sequence information from the Ensembl database (Ensembl Gene ID ENSG00000178403) and cloned under the control of the TRE tight (Tetracycline Response Element) promoter in a PiggyBac/Tet-ON all–in-one vector [14]. This vector contains a CAG rtTA16 cassette allowing constitutive expression of the Tet-ON system and an Hsv-tkNeo cassette for generation of stable IPS clones. After trypsinization into single cells with Tryple express reagent (ThermoFisher), approximately 1 × 10^6^ iPS cells were nucleofected by an Amaxa nuclefector device using Human Stem Cell Nucleofector Kit 1 (Lonza, Basel, Switzerland) and Program B-016 with 4 µg of Ngn2 plasmid and 1 µg of the dual helper plasmid. Subsequently, cells were replated on Matrigel plates with mTeSR medium containing 10 µM of Rock inhibitor. Antibiotic selection (G418 0.1 mg/mL) was applied 48 h later. Stable clones appeared within 1 week. To differentiate iNgn2 neurons, 1 × 106 of iPS cells were plated on a 6 cm Matrigel plate in proliferation medium (DMEM/F12 with Glutamax supplemented with 2% B27 and 1% N2, 10 ng/mL hEGF, 10 ng/mL hFGF, 1% Pen/Strep (all from ThermoFisher) containing Rock inhibitor (10 µM) for 1 d and doxycycline (1 ug/mL) for 3 d. Three days later, cell-induced neurons were frozen down. Cells were thawed and recovered in differentiation medium for 24 h before transfection (Neurobasal supplemented with 2% B27, 1% N_2_, Pen/Strep, 1 mM Sodium Pyruvate, plus 10 ng/mL BDNF, GDNF and hNT3). To ensure the cells were post-mitotic at the time of electroporation, the cultures were treated with 10 µM cytosine arabinoside 24 h before and after.

Amplicon sequencing. Total genomic DNA was isolated from transfected cells using a DNA extraction kit (Qiagen, DNeasy Blood and Tissue kit, Germantown, MD, USA). The region of theoretical insertion was amplified by PCR using primers hnRNPA2B1 fw (5′-tcccgtgcggaggtgctcctcgcag) and hnRNPA2B1 re (5′-agctccgcagcctcgctcacgagg). A 500 bp fragment could be obtained after 40 amplification cycles with denaturation at 95 °C for 30 s, annealing at 60 °C for 30 s and extension at 68 °C for 45 s using a proofreading Accuprime Pfx Mix (ThermoFisher). After gel purification, the PCR fragments were used to prepare sequencing libraries with the Ovation Low complexity Sequencing System (NuGEN, cat.9092-256). Libraries were sequenced in paired-end mode, at a read length of 2 × 300 bp, using the MiSeq platform (Illumina, San Diego, CA, USA). We obtained 10,253,514 and 11,140,907 read pairs for the control and the CRISPR samples, respectively. Read quality was assessed by running FastQC (version 0.10) (Babraham Institute, Cambridge, UK) on the FASTQ files. Since the sequencing quality decreased continuously after base 200, we trimmed the reads to a length of 240 base pairs before alignment. The FASTQ files were aligned against an extended human reference genome (based on GRCh38) using BWA version 0.7.15 with default parameter settings [15]. The extension of the genome consisted of the addition of two artificial chromosomes which contained the insert in forward and reverse orientations concatenated with 220 bp of chromosome 7 before base 26,200,579 and 220 bp after base 26,200,580 on chromosome 7. The alignments were then processed by a custom script to compute the different alignment classes of fragments.

Western blot analysis. At 72 h after transfection, cells were washed with PBS and lysed in RIPA buffer (ThermoFisher) containing a protease inhibitor cocktail (Roche). Protein concentrations in the samples were determined using a colorimetric assay based on the Lowry method (DC Protein Assay, Biorad) following the manufacturer’s instructions. Equal amounts (ranging from 5 to 10 µg) of total protein extracts were loaded after a 10 mn denaturation step at 70 °C on Nu-PAGE 4–12% Bis-Tris Gels (ThermoFisher) and separated by electrophoresis using MOPS buffer under reducing conditions. Proteins were transferred on a 0.2 µm PVDF membrane by electroblotting (TransBlot Turbo system, BioRad). After one hour of room-temperature blocking of the membrane in Odyssey PBS-Blocking buffer (LI-COR), the commercial primary antibodies anti-V5 (ThermoFisher), anti-Vimentin (Abcam) and anti hnRNPA2B1 (Abcam) were diluted 1:2000 in fresh blocking buffer, added on the membrane and incubated overnight at 4 °C. Membranes were washed twice in TBS buffer containing 0.1% Tween 20 (TBST) and incubated with fluorescence-labeled secondary antibodies (Goat anti-Mouse IR Dye 680 RD, or Goat anti-Rabbit IR Dye 800CW, LI-COR) diluted 1:10,000 for 1 h at room temperature. After 3 wash steps in TBST, the labeled proteins were visualized using an Odyssey Infrared Imaging system (LI-COR, Lincoln, CA, USA).

Immunostaining, imaging and FACS sorting. Immunostaining was performed according to standard protocols. Briefly, 72 h after transfection, cells grown on 96-well plates were fixed and permeabilized for 20 min at 4 °C with Fixation/Permeabilization solution (BD Biosciences). Cells were washed 3 times in PBS and then blocked in PBS containing 5% donkey serum, 1% bovine serum albumin (BSA) and 10% Perm/Wash buffer (BD Biosciences) at RT for 1 h. For V5 tag staining, cells were incubated at RT for 4 h with Alexa647-conjugated anti-V5 antibody, followed by rinsing 3 times in PBS before imaging. For other antigens, cells were incubated overnight at 4 °C with primary antibodies as indicated (Appendix A), washed three times with PBS and then incubated with 1:1000 Alexa Fluor-conjugated secondary antibodies (ThermoFisher) for 1 h at RT. The nuclei were visualized by counterstaining with NucBlue (ThermoFisher). Finally, the cells were rinsed a further 3 times in PBS before reading the plate on the Operetta Imaging System (PerkinElmer). The image analyses were conducted with the built-in Harmony software (PerkinElmer, Waltham, MA, USA).

For live cell imaging of mScarlet-Actin/Vimentin, HeLa cells were cultured on a 96-well plate for 7 days after transfection. The nuclei were stained with NucBlue Live ReadyProbe (ThermoFisher) and the culture media were replaced with FluoroBrite DMEM (ThermoFisher) supplemented with 3% FCS. The images were taken on Operetta and analyzed with Harmony software.

To enrich mScarlet-positive cells, HeLa cells were subjected to FACS sorting by a standard protocol. Briefly, cells were trypsinized and resuspended in PBS without Ca^2+/^Mg^2+^, with 2% FCS and 2 mM EDTA to obtain the single cell solutions (1 × 106 cells/mL). Cells were sorted on BD FACSAria Fusion, with a 561 nm laser for excitation and a 610/20 nm filter for emission. Positive cells were collected in a 6-well plate within the normal HeLa cell growth medium. Once the cells were confluent, they were split in a 96-well plate in FluoroBrite DMEM plus 3% FCS. Images were collected on Cell Voyager 7000 (Yokogawa, Tokyo, Japan) over 16 h, with one image every 10 min. The movie was generated with ImageJ.

Reverse transcription quantitative PCR. Total RNA was extracted from rat primary Schwann cells at the indicated time points with RNeasy (Qiagen) according to the manufacturer’s instructions. RNA concentration was determined by using a Nanodrop and reverse transcription was performed with 300 ng of total RNA by the Reverse Transcription kit (Thermo Fisher Scientific, Waltham, MA, USA). Quantitative PCR (qPCR) reactions were carried out on a 7900 HT AB instrument with TaqMan universal buffer (Thermo Fisher Scientific) and primers master mix designed in-house and produced by Microsynth (Appendix A). mRNA levels were normalized to levels of PpiB and ActB mRNA (Taqman probe from Thermo Fisher Scientific) in each sample.

HiBiT Nano-Glo assays. A palindrome-like HiBiT sequence (Appendix A) was knocked into Oct-6 in rat SCs, and cells can be expanded for a further 5~6 passages before use. The cells were split in a 384-well plate with 10,000 cells per well. Cells were cultured for 3 days in FSK-free medium before FSK was added to the wells at various concentrations. The expression level of Oct-6-HiBiT was determined with the NanoGlo HiBiT lytic assay kit (Promega, N3040). Briefly, Nano-Glo substrate and LgBiT protein were mixed with the lytic buffer (1:50 and 1:100, respectively). The volume of culture media was adjusted to 30 µL each well and then an equal volume of reagents was added. After 10 min, luminescence signals were measured on an EnVision Plate Reader (PerkinElmer, Waltham, MA, USA).

A palindrome-like HiBiT sequence (Appendix A) was knocked into IFIT1 in SkMCs, and cells were expanded for 10 days in the growth medium. The cells were split in a 384-well plate with 5000 cells per well. After cells attached, Interferon-β (IF014, Merck, Kenilworth, NJ, USA) was added to the wells at various concentrations. The expression level of IFIT1-HiBiT was determined with the NanoGlo HiBiT lytic assay kit.

Statistical analysis. All statistical analyses were conducted using the GraphPad Prism. Data were expressed as the mean ± standard deviation. Unpaired *t*-test is used in Appendix A.

## 3. Results

To increase the chance of target integration, we limited the size of donor DNA to 100 bp and thus focused initially on peptide tags. First, we aimed to label the heterogeneous nuclear ribonucleoprotein A2/B1 (HnRNP) in HeLa cells with a V5 tag. A gRNA was designed to target the 5′ UTR (DSB at 2 bp before ATG) of HNRNPA2B1, and the dsDNA donor was synthesized as complementary oligonucleotides and annealed. To ensure that the V5 tag is translated in the frame, an ATG was added to its 5′ end, as well as 1 bp to the 3′ end (Figure 1a). Three days after transfection, knock-in efficiency was assessed by anti-V5 immunostaining: 40 ± 1.5% of the cells displayed strong V5 signals that exclusively localized in the nucleus (Figure 1b). To further confirm the correct insertion of the V5 tag into the HnRNP, we performed Western blotting to check the size of the tagged proteins: two bands were visualized by the V5 antibody, with their sizes corresponding to the predicted chimeras (Figure 1c). Although this result already suggested that the V5 sequence was integrated into HNRNPA2B1 as intended, we went on further to confirm its correct location at nucleotide precision by amplicon sequencing. 

To overcome the genomic instability issue of Hela cells, we used HCT116 cells for NGS-based analysis of targeting events (Appendix A). We classified the reads from the amplicon sequencing into different alignment categories: 3.8% of the fragments aligned to the wildtype genome and contained no indels, whereas 54.7% of the fragments aligned to the wildtype genome and contained at least one indel. A total of 16.5% of the fragments aligned to the genome sequence containing the forward orientation of the insert without an indel and 18.4% of the fragments aligned to the genome sequence containing the 46 bp reverse insert without an indel. The forward or reverse direction of the alignment was determined with respect to the orientation of the gene HNRPA2B1. A total of 6.5% of the fragments aligned to the genome sequence containing the 46 bp insert and at least one indel (Figure 1). The ratio of forward and reverse integrations of the V5 sequence is about 50:50, and only forward integration (16.5%) can result in a functional expression of the V5 tag. 

To calculate the functional V5 tagging at the cellular level, we use the following equation: ***C*** = {1 − (1 − ***g***/100)^2^} × 100, where ***g*** stands for the percentage of functional genome integration, which is 16.5 in this case; and ***C*** represents the percentage of functional V5 expression at the cellular level, which is 30.3. The number matches well with the immunostaining result (Appendix A), which suggests 31.3 ± 3.9% of HCT116 cells express V5-HnRNP.

To test HiTag’s efficiency with another target, we tagged Vimentin in HeLa cells. Not only was high knock-in efficiency (over 60%) achieved, but a clear overlap of V5 and Vimentin antibody staining was also observed (Appendix A). Furthermore, V5-tagged Vimentin could be recognized by both antibodies in the Western blot (Appendix A). Similarly, β-actin and a cytosolic/nucleoli protein (RLP10A) could also be tagged with the V5 epitope in HeLa cells (Appendix A). The method achieved similarly high efficiencies in other common cell lines (31.3~50.7%) (Appendix A). We also achieved positive knock-in in human induced pluripotent stem (iPS) with the HiTag method (Appendix A).

CRISPR knock-in is generally not applicable to post-mitotic cells since their HDR mechanism is inactive. However, by using iPS-derived iNgn2 neurons, we demonstrated that the HiTag approach also worked in these cells (Figure 2a). Primary cells can be problematic for CRISPR knock-in too, as they are not suitable for clonal selection. We extended the application of HiTag into primary cells: as shown in Figure 2b,c, Vimentin labeling could be achieved in 23.9 ± 0.6% of primary rat tenocytes, as well as 36.3 ± 2.5% of rat Schwann cells (SCs), while V5-tagged Sox10 was visualized in 9.9 ± 0.2% of cells (Appendix A). Surprisingly, the HiTag protocol was even more efficient in primary human skeleton muscle cells (SkMC): V5-Vimentin signals could be observed in 66.3 ± 2.5% of cells (Figure 2d).

Gene tagging has applications in cell biology far beyond just labeling a protein for immunostaining. For example, an endogenous protein can be tracked in live cells by tagging with fluorescent proteins [5]. We introduced a dsDNA fragment encoding mScarlet [16] into Vimentin or β-actin in HeLa cells via HiTag. Filament structures were observed in 0.6~1.5% of live cells after 7 days (Figure 3a), and mScarlet-positive cells were enriched by fluorescence-activated cell sorting. The subcellular location of mScarlet-Vimentin could then be tracked using time-lapse fluorescence imaging (Appendix A).

Other possible applications include the ability to modify subcellular protein localization, expression level or stability by knock-in signal peptides, promoters/Kozak sequence or degrons [17]. As a proof of concept, a myristoylation sequence was added to V5-HnRNP (Appendix A) to disrupt its nucleus location: while 96.9 ± 0.9% of control cells display V5 signals in nuclei, 71.1 ± 8.5% of signals were cytosolic or on cell membranes when the myristoylation peptide was integrated (Figure 3b). 

Given the high efficiency of the HiTag protocol, we wondered whether it was possible to develop functional assays in primary cells via CRISPR knock-in. Oct-6 is a key transcription factor in the myelination pathway in Schwann cells. Its mRNA expression is up-regulated upon stimulation of the cAMP pathway [18] (Figure 3c). To measure this up-regulation at the protein level, a HiBiT sequence (an 11-amino acid subunit of Nanoluc luciferase [19]) was attached to the *C*-terminus of the Oct-6 protein in rat primary Schwann cells by HiTag. After the cells were incubated with Forskolin (FSK), which increases the intracellular cAMP level, up-regulated Oct-6 protein expressions were evident in the Nanoluc assay (Figure 3c). 

To further validate this approach, the HiBiT sequence was attached to the *C*-terminus of the IFIT1 protein in human SkMC. After incubation with IFNβ, which induces IFIT1 expression [5], a dose-dependent response was observed in the luciferase assay (Figure 3d). Therefore, we were able to develop a high-throughput format assay to measure endogenous protein expression in primary cells via CRISPR knock-in.

## 4. Discussion

Here, we report HiTag as a simple and efficient protocol for the CRISPR knock-in method that can be applied to various targets in a wide range of cell types. Although its exact mechanism remains elusive, HiTag knock-in seems to involve DNA-dependent protein kinase (DNA-PK), as the DNA-PK inhibitor [20] abolished the knock-in effect. In contrast, SCR7, a commonly used NHEJ pathway inhibitor [21], did not block HiTag knock-in at all (Appendix A).

After cutting of the genomic DNA by Cas9, knock-in of dsDNA donors needs to compete with the self-ligation of free ends. It is conceivable that the more dsDNA is used in transfection, the higher the knock-in efficiency would be. Indeed, this is shown in Appendix A. In some cell types, we observed cell toxicity issues when 1 µM or higher concentrations of donor DNAs were used, especially when they are more than 100 bp in length (data not shown). An inverse correlation between knock-in efficiency and the length of dsDNA donors is also evident (Appendix A). The efficiency was reduced even further (around 1%) when we tried to knock-in fluorescent proteins such as mScarlet (Figure 3a). 

The knock-in efficiency leading to the expression of the desired tagged protein is probably also linked to the intrinsic properties of the cells: genome stability and accessibility, repair machinery accountable for appropriate integration and regulation of translation. We have observed more than 50% efficiency with V5 knock-in in HnRNP in HEK293 cells (Appendix A), but only 31% in HCT116 cells (Appendix A). Highlighted by our deep sequencing results, it is important to keep in mind that the efficiency of genome editing is not directly linked to accurate integration of dsDNA: the 96% modified HCT cells contained only 31% of the integration events, and amongst these, only half were in the needed orientation to get a tagged protein. These numbers will probably differ from one cell type to another and can result in a variety of experimental outcomes. Interestingly, once dsDNA donors were incorporated, the chance of precise integration (~35%) was far higher than that with an indel (6.5%). This suggests the intrinsic accuracy of the NHEJ mechanism [4,5,6,7], and the observed high indel frequency without donor integration is probably the result of forced errors with the Cas9 presence. 

Like other CRISPR knock-in methods, the successful application of HiTag depends on the targeting and cutting efficiency of the gRNA/Cas9 complex, and the detection of the tagged proteins heavily relies on their endogenous expression level. Issues such as off-target editing, monoallelic editing and indels will also confound the biological readout. Ideally, deep sequencing should be performed for each knock-in experiment. For cell lines, some of the problems could be addressed by the labor-intensive clone selection, whereas for primary cells, the potential impact of the “unwanted” editing needs to be considered when results are interpreted.

The HiTag approach also has its own intrinsic limitations: (1) The orientations of the integrated donor DNAs are non-controllable and might result in non-functional knock-in. If the tags are targeted to the *C*-terminus of a protein, this problem could be partially overcome by using palindromic sequences, as shown in Appendix A. However, use of longer palindromic donors would also reduce the knock-in efficiency. (2) Compared to HDR-based protocols, the HiTag method is more likely to result in “off-target” knock-in. Therefore, when designing their experiments, future users of the HiTag protocol should evaluate potential off-target sites of their sgRNA sequences with an off-target prediction program such as Cas-OFFinder (http://www.rgenome.net/cas-offinder/). (3) Compared to HDR-based knock-in protocols, HiTag is even more restricted by the availability of suitable gRNA binding sites for the insertion. For example, knock-in can be efficient within ± 10 bp around DSB sites for HDR-based protocols [22]; however, it can only happen at DSB sites for HiTag. For tagging the *N*-terminus of a protein, the insertion site could be either before or after ATG. If it is within the 5′ UTR, one must consider that an additional sequence (between the DSB site and the original ATG) will be attached to the 3′ end of dsDNA donors and might have unwanted consequences. For tagging at the *C*-terminus of a protein, the DSB site must be the 5′ of the stop codon. Knock-in might result in the loss of a few residues at the *C*-terminus (encoded by the sequence between the DSB site and the stop codon), although this could be compensated by designing a dsDNA donor containing the deleted genomic sequence (with a mutated gRNA binding site). (4) The delivery of RNP into cells via electroporation is essential for HiTag, thus its throughput is rather low in the current format. However, electroporation devices are available in 96-well format as the alternative. 

Despite being potentially less specific compared with some HDR knock-in methods [23,24], one apparent advantage of the HiTag method lies in its efficiency. While many published data have claimed high knock-in efficiency, the numbers quoted were obtained after selection or enrichment [5,25,26]. With the HiTag protocol, we have been able to achieve 30–60% efficiency (at the cellular level) without selection. This unselected cell population might actually be advantageous for certain experiments, as it minimizes some of the biases inherent to clone selection. However, if a uniform cell population is preferred, the high efficiency of HiTag might help facilitate the selection process.

The second benefit of our protocol lies in its ease and simplicity: while other NHEJ-based methods rely on the vector expression of gRNA and the Cas9 protein [5,6,25,26], which requires intensive and expensive laboratory work, all reagents used in HiTag methods are readily available from commercial suppliers. In particular, we realize that synthetic dsDNAs are far more cost-effective donors compared to PCR amplicons or plasmid fragments. Combined with the fact that clonal selection is not necessary with HiTag, experiments on target proteins could be finished within two weeks, from design to a functional test.

Lastly, the importance of our method also lies in its diverse application. Primary cells are generally not suitable for CRISPR knock-in. However, given the high efficiency of HiTag methods, we have established several high-throughput luciferase assays to monitor the expression of endogenous proteins in primary cells.

In summary, HiTag offers a highly efficient CRISPR knock-in strategy that can be quickly applied to many target genes and cell types. We expect the simplicity of this approach will encourage more studies to be performed on endogenous proteins in their native environments.

## Figures and Tables

**Figure 1 cells-09-02618-f001:**
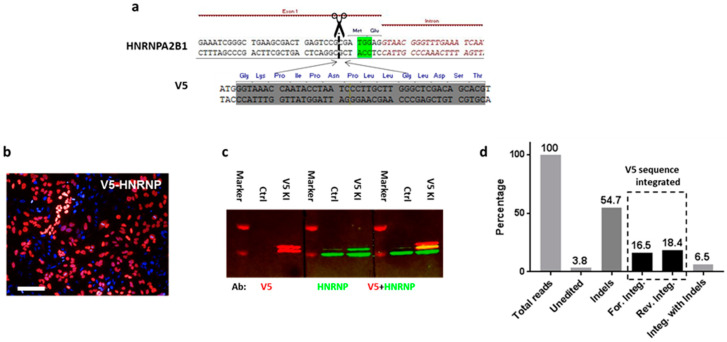
HiTag enables highly efficient CRISPR knock-in for imaging. (**a**) Sequences for HNRNPA2B1 (5′ end) with the PAM site (green) and cleavage site (dotted line); V5 sequence with ATG and an extra base at 3′. (**b**) Immunostaining with V5 tagged HnRNP in HeLa cells (*n* = 6). Scale bar = 100 µm. (**c**) Lysates from untransfected or V5-HnRNP knock-in HeLa cells were subjected to Western blotting. Both V5 (left) and HNRNP (middle) antibodies visualized two bands in Western blots, corresponding to A2/B1 isoforms. V5-labeled fragments are slightly larger than those recognized by the HnRNP antibody (right). (**d**) The ratio of forward and reverse integrations of the V5 sequence into the HnRNP genomic sequence is about 50:50, and only forward integration (16.5%) can result in a functional expression of the V5 tag.

**Figure 2 cells-09-02618-f002:**
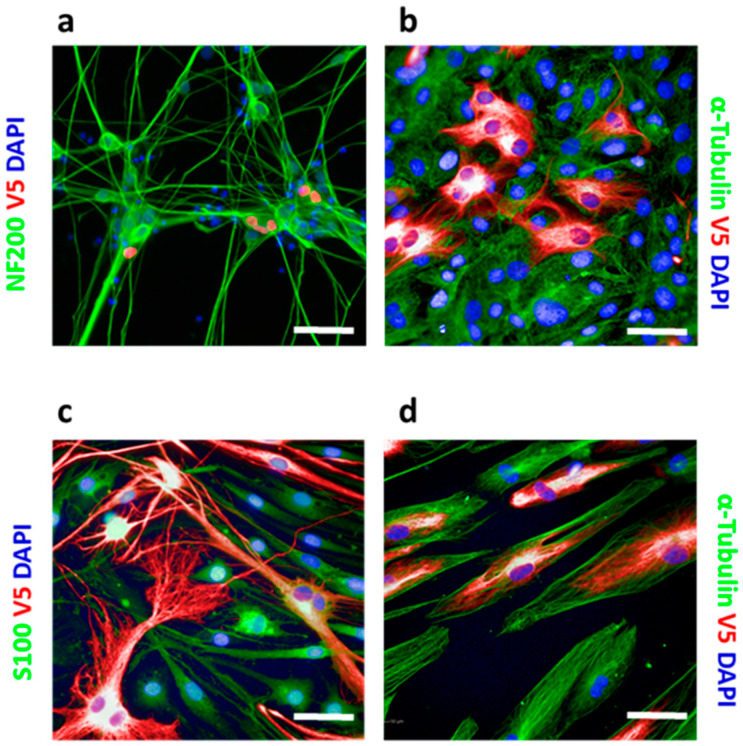
HiTag enables highly efficient CRISPR knock-in for imaging. (**a**) A total of 5.9 ± 1.6% of iNgn2 cells display V5-HNRNP (red) signals. Cells were co-stained with neuronal marker NF200 (green) (*n* = 4). (**b**) V5-Vimentin (red) in primary rat tenocytes, co-stained with α-Tubulin (green) (*n* = 10). (**c**) V5-Vimentin (red) in primary rat Schwann cells (SCs), co-stained with SC marker S100 (green) (*n* = 8). (**d**) V5 knock-in Vimentin (red) in primary human skeleton muscle cells, co-stained with α-Tubulin (green) (*n* = 4). Scale bar = 50 µm.

**Figure 3 cells-09-02618-f003:**
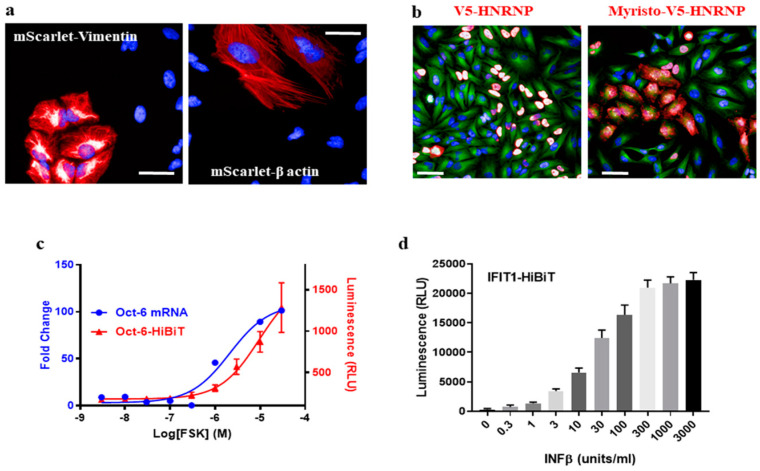
Multiple applications of HiTag in cell biology. (**a**) mScarlet was inserted into the *N*-terminus of Vimentin or the 5′ UTR of β-actin in HeLa cells. Live cell fluorescence images were taken 7 days after transfection (*n* = 10 each). Scale bar = 50 µm. (**b**) V5 HNRNP (left) and Myristoylated V5-HNRNP (right) were visualized by V5 antibody staining (red) in HELA cells. Addition of myristoylation sequence targets the endogenous protein to the cell membrane (*n* = 5). Cells were co- stained with α-Tubulin (green). Scale bar = 50 µm. (**c**) Primary rat SCs were incubated with FSK for 24 h, and Oct-6 mRNA expression was determined by qPCR (blue trace) (*n* = 3). To measure Oct-6 at the protein level, the HiBiT sequence was inserted at its *C*-terminus. The cells were treated with FSK at various concentrations and subjected to NanoLuc assay after 48 h (red trace) (*n* = 4). EC50 for FSK was 2.1 and 9.6 µM in qPCR and HiBiT assay, respectively. (**d**) IFIT1 was tagged with the HiBiT sequence at its *C*-terminus in primary human skeleton muscle cells (SkMC). The cells were treated with IFNβ at various concentrations and subjected to NanoLuc assay after 16 h (*n* = 10). The calculated EC50 for IFNβ was 27.5 unit/mL. The images shown are representative of the indicated number of separate experiments shown in brackets.

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
