# Peer review of "A Simple and Efficient CRISPR Technique for Protein Tagging"

_cells, 2020, doi:10.3390/cells9122618_

Round 1
Reviewer 1 Report
Genome engineering using CRISPR is quite useful in many research field. In this manuscript, the authors demonstrated endogenous protein tagging by targeted integration of the tag encoded sequences into the 5’end or 3’ end of the gene of interest. Instead of homology-directed repair (HDR), the authors used a non-homologous end-joining (NHEJ) pathway for targeted knock-in, named Homology independent gene Tagging (HiTag), which showed higher efficacy and flexibility for various cell lines including human primary cells. Although this concept has been introduced several times by previous studies so that this study has low novelty, this method is practical for broad users. I would raise several concerns that should be addressed before publication, as follows.
1) When applying HiTag, the endogenous target can have several different outcomes; unedited, indels without integration, forward integration of the dsDNA donor template (desired~!!), reverse integration of it, and integration with indels that may be caused by NHEJ pathway during integration or by cells’ unknown nucleases before integration. Further modification of donor DNA for protecting may decrease the case of ‘integration with indels’ in part.
In this regard, the high-throughput sequencing data of Figure S2 is quite valuable. In case that editing efficiency is very high (>96%), the desired outcome (For.Integ.) accounts for 16.5%. I suggest that this Figure S2 should be moved to main text and deeply discussed.
2) I am curious why the authors limited the length of dsDNA donor up to 100 bp. Although the efficiency can be reduced, I think that homology independent targeted integration (HITI) has no limitation of the donor’s length. Have the authors tried to integrate whole fluoresce proteins (FPs) such as GFP and RFP (~1 kbp) and compared the knock-in efficiency between small tag and FPs?
3) As described in the discussion part, the strategy of HiTag has fundamental limitations itself; i) the orientations of the dsDNA donor are non-controllable, ii) accompanies indels, iii) limited target number compared to HDR-mediated knock-in, iv) the addition of one amino acid when targeting 5’ UTR before ATG start codon.
In addition, I would raise an off-target knock-in issue, compared to HDR-mediated knock-in approach. It is well known that CRISPR can cause off-target DNA cleavage, but the HDR will not be occurred because there are no homology sequences at the off-targets. In contrast, HiTag donor can be easily integrated into the off-target cleavage site. Have the authors tried to detect this off-target integration at poteintial off-target sites? The off-target prediction program such as Cas-OFFinder (http://www.rgenome.net/cas-offinder/) will be helpful. It should be discussed.
4) (minor) The labels in Figures 1d-1g are hard to see due to their color. It is better to note them outside the figures.
Author Response
Dear reviewer:
1) Thank you very much for your suggestion. We have modified the figures accordingly and added a section in the discussions (page 8, lines 323-335)
2) As we showed in Fig. S9, the knock-in efficiency decreases rapidly once the donor size is over 100bp. We have also have tried the knock-in for fluorescent proteins, such as mScarlet. Data were shown in Fig. 3a. As predicted, the efficiency was much lower, around 1% (see page 7, line 268)
3) We thank the reviewer for the good question and agree to add “off-target knock-in issues” among the possible limitations of HiTag. In fact, as for the other CRISPR-based methods and depending on the quality of the sgRNA sequence used, HiTag may lead to off-target DNA cleavage and possibly to off-target knock-in issues. In the current work, we confirmed the “on-target” knock-in by immunostaining or functional assay, but we did not try to detect the presence of off-target sites. We have added one sentence to advice future users of the HiTag assay to evaluate potential off-target sites of newly design sgRNA sequences with an off-target prediction program such as Cas-OFFinder (http://www.rgenome.net/cas-offinder/) (page 8, lines 347-350)
4)Thank you for your suggestion. We have modified the figure as you suggested.
Reviewer 2 Report
The manuscript by Zhen et al et al reports a novel method for high efficiency integration of peptide tag coding sequences.
Integration of peptide tag coding sequences is a common goal of gene editing experiments and can be difficult to achieve. The novel method is therefore exciting and will be a very significant addition to available gene editing strategies. The experiments supporting the manuscript are carefully performed and the efficiency of the novel method is demonstrated in numerous cell types, including primary cells that are not generally amenable to gene editing. Limitations of the method are also discussed, which I appreciated.
Minor comments.
- many previous studies have failed to detect SCR7 activity in cultured cells and the negative results reported here should not be taken to suggest that ligase IV is not involved.
- the Western blot shown in Figure 1c is confusing. Could the authors provide some explanations to the observations here below?
- I would expect the HNRNP antibody to detect both V5-tagged HNRNP isoforms but that’s not the case. Only the smaller V5-tagged isoform is detected with the HNRNP antibody.
- based on the HNRNP antibody staining, one isoform appears to be much more abundant than the other. However, that’s not the case for the V5-tagged HNRNP isoforms. Both V5-tagged isoforms give bands of identical intensity.
- The authors report that tag insertion is much more efficient at lower Cas9 RNP concentrations. Checking indel rates in parallel would be important to better appreciate this intriguing observation.
Author Response
Dear Reviewer,
Thank you very much for your comments.
1) as you suggested, we have removed the statement about SCR7
2) Indeed the HNRNP antibody should detect four different bands: the two isoforms and their V5-tagged versions. We suspect the V5 tag insertion could have destroyed the epitope for HNRNP antibody, and therefore, only those two non-tagged isoforms were detected by the HNRNP antibody.
However, the western blot result is much clearer for Vimentin knock-in (supplementary Fig.3b), where two bands were observed with Vimentin antibody staining, but only one corresponding band with V5 staining.
3) Thank you for pointing this out. We think it might be because of different maturation time for two isoforms. HnRNP antibody staining represents total HnRNP protein available, while V5 staining only reveal the de novo V5 tag proteins after knock-in. We perform the staining 72 hrs after electroporation, and one isoform might mature faster, thus show higher expression at that time point. However, we tried to avoid this discussion in the manuscript as the biology of HnRNP is outside the scope of our study
4) Thank you for your comments. We were surprised by this result and agree that checking indel rates in parallel will be ideal to understand this observation. However, we have since realized this might be caused by the contaminating endotoxin within our purified Cas9 protein. Therefore, we decide to remove the figure to avoid confusion.
Reviewer 3 Report
The authors explore HiTag, a HDR independent method. Interesting concept, worth pursuing. One of the main issues is the lack of specificity at which the integration can potentially happen. Homology directed repair ensures that the repair happens at a low level, but at the region we target.
With dsDNA and no homology arms, wherever the guide creates DSBs, including on and off targets, there is a potential for integration.
There is no insights into the mechanism, but the study offers only observations
Include additional references
If authors can provide mechanistic insights and identify off-target integration, it would be a useful study
Author Response
Dear Reviewer,
Thanks you very much for your comments. While we acknowledge the accuracy / specificity of HiTag protocol might be inferior to HDR based approaches, its advantages are also evident: efficiency and simplicity. We envision this method can be widely applicable for cell line generation, where most of its accuracy issue can be resolved with single cell cloning.
It is clear that HiTag knock-in involves canonical non-homologous end joining (c-NHEJ) pathway, and DNA-dependent protein kinase also plays an essential role (our data). However, the actual mechanism of NHEJ depended Crispr knock-in remains largely elusive, and requires the collaborative efforts from science community to resolve.
Round 2
Reviewer 1 Report
The authors have mostly answered the issues I raised in the earlier review. I would recommend publication of this revised version.